# Yeast-Fermented Rapeseed Meal Extract Is Able to Reduce Inflammation and Oxidative Stress Caused by *Escherichia coli* Lipopolysaccharides and to Replace ZnO in Caco-2/HTX29 Co-Culture Cells

**DOI:** 10.3390/ijms231911640

**Published:** 2022-10-01

**Authors:** Ionelia Taranu, Gina Cecilia Pistol, Andrei Cristian Anghel, Daniela Marin, Cristina Bulgaru

**Affiliations:** Laboratory of Animal Biology, National Institute for Research and Development for Biology and Animal Nutrition, Calea Bucuresti No. 1, Balotesti, 077015 Ilfov, Romania

**Keywords:** ZnO, fermented rapeseed meal, co-culture cells, inflammation, oxidative stress, signaling molecules, in vitro

## Abstract

(1) The present study tested in vitro the capacity of a fermented rapeseed meal extract to reduce medicinal ZnO, which will be banned at the EU level from 2023 onwards because of its potential to cause environmental pollution and the development of Zn resistance in gut bacteria. Rapeseed meal could be an important ZnO substitute as it has antioxidant/radical scavenging properties due to its content of bioactive compounds (e.g., polyphenols). (2) Protein array and flow cytometry were used to detect apoptosis, oxidative stress production, and inflammatory and signaling-related molecules in Caco-2 and goblet HT29-MTX co-culture cells challenged with Escherichia coli lipopolysaccharides and treated with ZnO and FRSM. (3) LPS induced cell death (21.1% vs. 12.7% in control, *p* < 0.005); apoptosis (16.6%); ROS production; and overexpression of biomarkers related to inflammation (63.15% cytokines and 66.67% chemokines), oxidative stress, and signaling proteins when compared to untreated cells. ZnO was effective in counteracting the effect of LPS, and 73.68% cytokines and 91.67% of chemokines were recovered. FRSM was better at restoring normal protein expression for 78.94% of cytokines, 91.67% of chemokines, and 61.11% of signaling molecules. FRSM was able to mitigate negative effects of LPS and might be an alternative to ZnO in pig diets.

## 1. Introduction

The health of the gut in piglets is crucial during weaning, the most difficult period in the pig’s life during which the intestinal epithelium is susceptible to inflammation caused by different pathogens like *Escherichia coli*, *Salmonella*, *Rotavirus*, etc. Diarrhea is one of the consequences of gut inflammation and pathogenic toxins. Zinc oxide (ZnO) in concentrations higher than 1000 ppm is licensed as a veterinary medicinal product for the treatment and control of post-weaning diarrhea in swine [1]. Its importance has increased since the banning of in-feed antibiotics [2]. At the intestinal level, ZnO maintains epithelial integrity and function, inhibits inflammation, and improves mucosal repair and cellular permeability [3,4]. However, despite the widespread usage of ZnO indicating its effectiveness in controlling post-weaning crises, its use as a medicinal additive (mainly at pharmacological dosages of 2500 to 3000 ppm) will be banned from 2023 in the EU, because it has the potential to cause environmental pollution and the development of intestinal bacteria resistance to Zn [5]. Some European countries have already banned the therapeutic use of zinc oxide in pig diets, while it is still used in others. Consequently, it is necessary to find new alternative compounds to in-feed ZnO or antibiotics that can produce similar effects without the risks of environmental problems and bacterial resistance. This has opened up many opportunities for animal nutrition research. Alternative active compounds have been identified in plant extracts, different waste/byproducts, probiotics, organic acidifiers, and antioxidants [6]. Among the most studied and used probiotics are lactic acid bacteria (*Lactobacillus*, *Bifidobacterium*, *Enterococcus*, etc.) and yeasts (*Saccharomyces* sp.), which are able to prevent intestinal pathogen bacterial adhesion, to reduce the pH, and to stimulate the production of immune effector molecules or inhibitory substances, bacteriocins, organic acids, etc. [7,8]. *Saccharomyces* sp. contain in their cell walls bioactive polysaccharides like β-glucans which have anti-inflammatory and immunomodulatory properties in pigs [7,9,10].

Agro-industrial wastes and byproducts are also natural sources rich in bioactives known for their anti-inflammatory, antioxidant, and antimicrobial effects [11]. The very recent study reported in [12] which tested on in vitro Caco-2 cells the antioxidant potential and the effect on epithelial integrity of nine plant extracts and essential oils (grape seed extract, green tea extract, olive extract, ginger essential oil) proved these effects. All the tested botanicals were able to reduce the negative effects of oxidative stress and to improve epithelial integrity in different concentrations due to the antioxidant function of phenolic compounds. Additionally, Hao et al. (2015) [13] reported that dietary grape seed procyanidin supplementation ameliorated immune and antioxidant response as well as diarrhea incidence in weaned piglets by enhancing both serum IgM and IgG and the cytokine IL-2, which has an essential role in the induction of B- and T-cell differentiation [13]. Administration of grape seed and grape marc in the diet (1%) of piglets was able to decrease the gene expression of pro-inflammatory cytokines IL-6, TNF-α, IL-8, and IL-1β in the duodenum [14]. A positive effect of some byproducts rich in polyunsaturated fatty acids (PUFAs) on antioxidant status via an increase of total antioxidant capacity and the activity of several important antioxidant enzymes (CAT, SOD, GpX) was also found by [15,16,17]. The activation of nuclear factor erythroid 2–related factor 2 (Nrf2) and the inhibition of nuclear factor kappa-light-chain-enhancer of activated B cells (NF-kB), responsible for induction of inflammatory response, are among the molecular mechanisms related to PUFAs. The intake of PUFAs changed the Th1/Th2 cytokine balance by inhibiting Th1-type cytokines with the decrease of IL-1β, IL-6, TNF-α, and IL-8, cytokines involved in inflammation [18].

Although some agro-industrial waste are rich in bioactive compounds with antimicrobial and anti-inflammatory properties, they contain antinutritional factors which limit their use in animal feed. This is the case for rapeseed meal, which has a high glucosinolate content. Studies over the last decade have shown that solid fermentation, especially with lactic acid bacteria but also with some yeasts, significantly decreases the concentrations of these antinutrients [19,20]. Furthermore, microbiological fermentation generates new health-promoting bioactive compounds including lactate, butyrate, beta glucans, polyphenols, carbohydrate-degrading enzymes, and antimicrobial peptides, which play essential roles in modulating inflammation in the intestinal mucosa [21,22,23,24].

The fermentation process also improves the digestibility of feed nutrients [25]. The use of yeast-fermented rapeseed meal in pig after weaning has not been tested. The fermentation of rapeseed meal used in pig has been done mainly with lactic acid bacteria, and there have been very few studies investigating fermentation with other microorganisms, such as yeasts [23,26].

In this context, the present study included preliminary in vitro testing of the bioactive compounds from an extract of rapeseed meal fermented with *Saccharomyces boulardi* (*var. CNCM I0745*, FRSM), in order to bring supporting data for the inclusion of this byproduct as a substitute for ZnO in the feed of piglets after weaning. The comparison with ZnO was evaluated through the quantification of several inflammatory and oxidative stress mediators, as well as their responsible triggering mechanisms, in a complex intestinal co-culture cellular model including a monolayer of enterocyte Caco-2 and goblet HT29-MTX cells challenged with *Escherichia coli* lipopolysaccharides (LPS), as this is one of the most common intestinal pathogens.

## 2. Results

### 2.1. Effect of ZnO and FRSM Extract on Apoptosis in Caco-2/HT29MTX Co-Culture Challenged with LPS

The results presented in Figure 1 show that neither ZnO nor FRSM alone had any effects on apoptosis profile in Caco-2/HT-29MTX co-cultured cells. The addition of 1 µg/mL of LPS induced an increase of total dead cell percentage compared to control cells (21.1% vs. 12.7% in control, *p* < 0.005). Of the dead cells, apoptotic cells accounted for a higher percentage in LPS-treated cells (16.6%) compared to untreated cells (9.2%) (*p* < 0.005); the surface phosphatidylserine expression (as measured by the Muse Annexin V and Dead Cell Assay) demonstrated that 7.9% of cells were positive for this early-apoptosis marker in LPS-challenged cells, in contrast with only 4.3% in the control (*p* < 0.050, Figure 1). When LPS was added to the cells already treated with ZnO and FRSM, the effect of the LPS was diminished. The number of apoptotic cells in both phases was reduced, and these cells showed similar histograms to control cells and also to the cells treated with ZnO and FRSM alone (Figure 1).

### 2.2. Effect of ZnO and FRSM Extract on Oxidative Stress in Co-Cultured Caco-2/HT29MTX Challenged with LPS

#### 2.2.1. ROS Production

Exposure of co-cultured cells to 1 µg/mL of LPS significantly increased the percentage of ROS (+) cells compared to control cells (+57.40%, *p* < 0.005, Figure 2) and reduced that of ROS (−) cells (+8.71, *p* < 0.005, Figure 2). ZnO alone had no effects on either ROS (+) or ROS (−) cells (87.41% ROS (−) cells and 12.59% ROS (+) cells vs. 86.83% ROS (−) cells and 13.17% ROS (+) cells in control, Figure 2). In contrast, FRSM alone reduced the percentage of ROS (+) cells (+28.71%, *p* < 0.050 vs. control, Figure 2) and increased the percentage of ROS (−) cells (+4.4%, *p* < 0.050 vs. control, Figure 2). ZnO was much less effective than FRSM extract in counteracting the effect of LPS on ROS (+) cells (18.87%, *p* = 0.086 vs. 20.73% LPS, Figure 2). FRSM prevented the effect of LPS by reducing ROS (+) cells to below the control level (−53.21%, Figure 2).

#### 2.2.2. Protein Oxidation

The analysis of the oxidized (carbonylated) proteins in co-cultured cells presented in Figure 3 showed that the ZnO and FRSM treatment alone did not affect the protein carbonyl content when compared to untreated cells. In contrast, the LPS treatment produced the oxidation of cellular proteins, leading to a significant increase in protein carbonyl content in co-cultured cells (+218.52%, *p* < 0.001 vs. control, Figure 3). Similarly to the other oxidative markers, ZnO and FRSM treatments were able to counteract the effect of LPS by decreasing the concentration of protein carbonyl, with 70.93% (ZnO treatment) and with 67.44% (FRSM extract treatment) of the control level, respectively.

#### 2.2.3. Lipid Peroxidation

To evaluate the effects of ZnO and FRSM on lipid peroxidation, the levels of thiobarbituric-acid-reactive substances (TBARS) were detected in co-culture supernatants. As shown in Figure 4, the LPS challenge leaded to a significant intensification of TBARS production in Caco-2/HT29-MTX co-cultured cells (2.1-fold increase, *p* = 0.001 vs. control). The treatments with ZnO alone had no effect on TBARS concentration compared to control (1.3-fold increase, *p* > 0.05 vs. control), while FRSM alone induced a 1.7-fold increase of TBARS levels over the control (*p* = 0.043 vs. control). When LPS was added to the cells that had been already treated with ZnO and FRSM for 4 h, the concentration of TBARS decreased to below the level of that produced by LPS. (Figure 4).

### 2.3. Evaluation of the Effect of ZnO and FRSM Treatments on Global Protein Expression Profile upon LPS Exposure (Overview)

Exposure of co-cultured cells to LPS produced overexpression of 44 (65.67%) of the total (67) analyzed proteins involved in inflammation, oxidative stress, and signaling pathway, 12 being cytokines and 8 chemokines when compared to untreated cells (Table 1). The expression levels of 22 proteins were not affected by LPS. Overexpression of 24/36 signaling molecules was also produced by LPS in Caco-2-HTX29 co-culture in comparison to control cells.

Known as an anti-inflammatory product, ZnO was associated with overexpression of only 13 proteins (19.40%) belonging to the signaling protein group. This treatment did not result in overexpression of either cytokine or chemokine proteins. Expression levels of 45/67 (67.17%) proteins were decreased. Fourteen cytokines and elven chemokines that were overexpressed by LPS returned to the level of the control (Table 1). Additionally, ZnO decreased toward the level of control the expression levels of 20 signaling molecules and increased those associated with transcription factors/cell proliferation/cell differentiation. Similar results were obtained for FRSM alone, suggesting that the fermentation process conferred anti-inflammatory properties to rapeseed meal (Table 1).

The two products were also effective when LPS was added to the co-culture after 4 h. In total, levels of 11 (16.42% of total proteins) cytokines, 9 (13.43% of total proteins) chemokines, and 27 (40.30%) signaling proteins overexpressed by LPS were recovered by ZnO (ZnO + LPS) treatment. FRSM extract in FRSM + LPS treatment showed a better ability to restore toward the control level protein expression for 14 (20.90%) cytokines and 11 of chemokines (16.41%, Table 1) associated with inflammation. However, some of these proteins (9) decreased to below the control level.

### 2.4. Effect of ZnO and FRSM Treatments on Protein Expression of Biomarkers Associated with Inflammation upon LPS Exposure

Expression of 31 proteins related to inflammation and oxidative stress (19 cytokines and 12 chemokines) was investigated in this study. Depending on the relationship with inflammation response and based on the classification described by [27], cytokines were divided into pro-inflammatory (12 proteins), anti-inflammatory (3 proteins), and inflammation-regulatory (4 proteins) functional groups.

As expected, 63.15% (12/19) of cytokines associated with inflammation were significantly (*p* < 0.05) overexpressed by LPS. Granulocyte colony-stimulating factor (G-CSF), strongly related to neutrophils, and tumor necrosis factor alpha (TNF-α), both powerful pro-inflammatory cytokines, were the most significant overexpressed proteins (+93.1% increase, *p* = 0.032 and +65.0% increase, *p* = 0.017 respectively) compared to control (Table 2). ZnO alone significantly decreased the expression of IL-13 protein (−47.8%, *p* < 0.05) when compared to control. In contrast, the action of LPS added into co-cultured cells that had already been treated for 4 h with ZnO and FRSM extract (ZnO + LPS, FRSM + LPS) was decreased, and 14/19 and 15/19 of inflammation-related proteins returned toward the control level. LPS also amplified the majority of chemokine proteins (8/12, 66.67%). Among these were MCP-1, with a +53.8% increase, *p* < 0.017, RANTES, with a +47.1% increase, *p* < 0.023; and others (Table 3). As in the case of cytokines, treatments with ZnO and FRSM were effective in counteracting the effect of LPS. Expression of 83.33% of analysed chemokines regained the control level in FRSM + LPS treatment and only 75.00% in the case of ZnO + LPS treatment. Their effect alone was similar with that of the control.

### 2.5. Effect of ZnO and FRSM Treatments on Metalloproteinase Activity upon LPS Exposure

Gelatine zymography analysis of co-cultured cell supernatants revealed the gelatinolytic activities corresponding to MMP-2 and MMP-9. The results presented in Figure 5 show that both ZnO and FRSM alone had no effects on MMPs’ activities in Caco-2/HT-29MTX co-cultured cells when compared to control cells. In contrast, LPS treatment led to a significant increase of both MMP-2 and MMP-9 activities over the level of control cells (+153% increase, *p* < 0.005 and respectively +20% increase, *p* < 0.050 vs. control, Figure 5). Addition of LPS in co-cultured Caco-2/HT-29MTX cells treated for 4 h with ZnO and FRSM registered a diminished effect, with the activity of MMP-2 and MMP-9 decreasing significantly (Figure 5).

### 2.6. Effect of ZnO and FRSM Treatments on Protein Expression of Biomarkers Linked to Signaling Pathway upon LPS Exposure

As expected, multiple very important signaling proteins associated with inflammation and oxidative stress pathways, such as MAPK-p38-α (+126.2% increase), CREB (+93.8% increase), c-Jun (+175.7 increase), and JNK (+58.4%), were significantly increased in co-culture in which LPS acted alone. *E. coli* LPS endotoxin also increased the expression of several transcription factors belonging to the signal transducer and activator of transcription (STAT) protein family (STAT2, +103.9% increase, STAT3, +141.3% increase) as well as that of proteins involved in cellular migration and cell proliferation/cell cycle/cell differentiation (PYK2, PLC-γ1, p53, Akt; Table 4A,B) which could have different consequences on cells. For example, the overexpression of p70S6k-T389 (+74.3%) is seen in some cancer cell lines [28,29], while Chk-2 (+74.3%) is increased in apoptosis [30]. Treatments of intestinal cells with ZnO and FRSM extract before LPS addition were able to counteract the effect of LPS exposure on signaling proteins linked to inflammation and to maintain the expression of cell proliferation/cell cycle/cell differentiation and transcription factors proteins close to those of the control (15/23 and 13/23, Table 4B). The decrease was more pronounced for certain proteins in the case of FRSM, even to below the control level.

## 3. Discussion

ZnO in pharmacological doses is widely used in pig nutrition during weaning to relieve intestinal inflammation and diarrhea caused by common pathogens such as *E. coli*, *Salmonella*, etc. Weaning is the major critical and healthcare period in pig production [3]. Zinc also strengthens the immune system. Due to suspicions related to bacterial resistance, environmental pollution, and potential toxicity, the use of zinc oxide in animal feed will be banned from 2023 in the EU. Little is known about the effect of ZnO on the gastro-intestinal tract. Alternative nutritional solutions for replacing ZnO have been tested and others still need to be tested. For this reason, in the present study we evaluated *Saccharomyces boulardi* fermented rapeseed meal as an alternative to ZnO and investigated its capacity to counteract the negative effects produced by *E. coli* LPS in an intestinal co-culture complex system including Caco2 cells as a model for the epithelial barrier and HT29-MTX goblet cells, which secrete mucin glycoproteins and form the mucus layer that limits the entrance of pathogens into the gut [31]. In this system, *E. coli* LPS caused increases in ROS production, TBARS and protein carbonyl concentration, and markers of protein and lipid oxidation. In contrast, co-cultures treated with ZnO and FRSM extract showed reduced levels of these markers compared to those of the control. Interestingly, the ability of FRSM extract to reduce ROS levels was much higher than that of ZnO, providing the first results to suggest that fermented rapeseed meal could be a viable alternative to ZnO. Mechanistically, the abilities of the two products to counteract oxidative processes are different. In the case of ZnO, long-term exposure leads to the induction of antioxidants such as metallothioneins on one side, and on the other side either the protection of protein group sulfhydryls or the reduction of (*) OH formation from hydrogen peroxide (H_2_O_2_) through the antagonism of redox-active transition metals, iron, and copper [32]. As for FRSM, the mechanism might be related to the fermentation process which enhance rape seed meal in bioactive compounds such as enzymes, vitamins and minerals (important co-factors) or polysaccharides with antioxidant properties. β-glucans, for example, are polysaccharides from the yeast wall with a high antioxidant activity against free radicals [33,34]. It was shown that local and systemic administration of β-glucans in rats reversed the increase of MDA, a marker of lipid peroxidation, and reduced GSH enzyme levels [35]. A protective effect of β-glucan against oxidative damage, similar to that of silymarin in uranyl-induced cytotoxicity damage, was also observed in vitro in hamster lung fibroblasts [36]. Bioactive compounds like polyphenols or polyunsaturated fatty acids from rapeseed meal, which are known for their antioxidant activity, are involved in chelating transition metals which contribute to free radical production [36,37]. These authors reported that polyphenols from red wine counteract the accumulation of hydrogen peroxide by decreasing TBARS level and superoxide dismutase activity in excess in zinc-deficient rats.

Oxidative stress is highly linked to inflammatory response, a process in which cytokines and chemokines play a fundamental role [4]. Indeed, in the present study, the challenge of co-cultured cells with LPS significantly increased the levels of pro-inflammatory cytokines and chemokines as well as the activity of pro-inflammatory metalloproteinases (MMP-2 and -9). Among these, the highest levels were reached by G-CSF (+93.1%), TNF-α (+65.0%), IL-1β (+41.8%), and IL-1α (+42.9%), the first involved in neutrophil survival, proliferation, differentiation and function, and the others being powerful pro-inflammatory cytokines known for their involvement in intestinal damage [38]. A high increase in IL-3, IL-7, and IL-5, all regulatory cytokines of inflammation, was also found. TNF-α is able to increase the capacity of epithelial cells to produce chemokines, which are very important chemoattractant molecules for neutrophil and monocyte/macrophage movement towards inflammatory sites and their infiltration in intestine [39]. For example, we found that MCP1 (+52.8%) and RANTES (+47.1%), specific recruiters for monocytes and leukocytes (eosinophils, and basophils, monocytes, natural-killer (NK) cells, dendritic cells) to sites of inflammation, were the most increased chemokines in the LPS- challenged Caco-2/HT29-MTX system. As in the case of oxidative stress, both ZnO and FRSM treatments were able to restore the levels of inflammatory mediators toward the control level. A beneficial capacity of ZnO in the protection of intestinal Caco-2 cells against inflammation induced by *Escherichia coli* (ETEC, strain K88) was also reported by [4], who showed that in vitro 0.2 or 1 mmol/L ZnO mitigated the increased expression of TNF-α, GRO-α, and IL-8. The mechanism of ZnO was to block the adhesion and internalization of bacteria and prevent cell permeability [4]. Along the same lines, results obtained in vivo on broilers or piglets indicated that ZnO attenuated pro-inflammatory cytokines (TNF-α, IL-1β, IL-6, and IL-8) in different tissues, intestine among them [3,40,41]. By decreasing IL-8, one of the strongest cellular chemoattractants, ZnO may reduce the number of degranulated neutrophils and thus decrease oxidative stress, inflammation, and intestinal damage [41]. It was shown previously that TNF-α and some other pro-inflammatory-cytokines are able to induce apoptosis in epithelial cells by increasing cellular permeability, with TNF-α acting as a trigger for apoptosis extrinsic pathway [42]. As shown in our study, LPS induced an increase in TNF-α protein expression and raised the percentage of apoptotic cells in both early and late apoptotic phases while in co-culture, and addition of ZnO and FRSM extract resulted in the number of apoptotic cells decreasing and reaching the level of control. During disease pathogenesis (inflammatory diseases for example), TNF-α also leads to increased cellular permeability by enhancing the transcription of myosin light chain kinase (MLCK) and occludin endocytosis [43]. In the present study, ZnO and FRSM prevented LPS-induced TNF-α. An important finding in the present study was that ZnO had a more pronounced anti-inflammatory effect than FRSM extract. It seems that the effect of ZnO on inflammation is controversial and depends on its form of administration [41]. Lei et al. [44], showed that serum IL-6 and TNF-α decreased at 72 h post-challenge with *Escherichia coli* K88 in piglets treated with conventional ZnO 2500 ppm and coated ZnO 1000 ppm (lipid matrix containing 40% ZnO and 60% hydrogenated palm oil). Another study [45] also found that coated ZnO at 380 or 570 mg Zn/kg improved villus height and crypt depth, gene expression of tight junction proteins, and IL-10, TGF-β1, and IgA concentration in the jejunal mucosa. In contrast, there are studies showing that ZnO can generate ROS and induce a pro-inflammatory effect, especially when it is in the form of ZnO nanoparticles [46,47].

Based on a protein array analysis of 36 inducible signaling proteins implicated in inflammation, apoptosis, and oxidative stress, we evaluated the underlying mechanism triggered by ZnO or FRSM extract alone or with challenge with *E. coli* LPS. We found that the majority of proteins implicated in inflammation and the transmission of apoptotic signal [42], such as CREB, c-Jun or JNK, MAPK-p38, and NF-kB, were overexpressed by LPS. Both ZnO and FRSM extract supplements were effective in reducing the increased expression of these intestinal-associated signaling molecules. The beneficial effects of ZnO and FRSM have already been shown in other studies. Che et al. [48] observed that dietary live yeast supplementation (*Saccharomyces cerevisiae* strain CNCM I-4407) decreased the gene expression of TLR4 and NF-kB in the mesenteric lymph nodes of piglets after *Escherichia coli* K88 challenge similar to a diet supplemented with an antibiotic and ZnO. Taranu et al. [49] also reported an anti-inflammatory protective effect on the intestine of a diet including rape seed meal fermented with *S. cerevisiae* by decreasing the protein expression of NF-kB and increasing that of Nrf2 in piglets after weaning. ZnO also decreased the expression of TLR4 and its downstream signals, MyD88, IL-1 receptor-associated kinase 1, TNF-α receptor-associated factor 6, and NF-kB genes with a simultaneous decrease of pro-inflammatory cytokines and chemokines in piglets fed 2200 mg Zn/kg from ZnO for one week [50]. However, it seems that the mechanism of ZnO action depends on the type of ZnO. Senapati et al. (2015) demonstrated that ZnO in form of nanoparticles enhanced the inflammatory response by activating the redox-sensitive NF-κB and MAPK signaling pathways [46].

## 4. Conclusions

In summary, our results confirmed the positive effects of ZnO against *E. coli* LPS, exerted by inhibition of ROS production, apoptosis, and pro-inflammatory cytokines and chemokines. Similar results were obtained with FRSM extract treatment, suggesting that it might replace commercial ZnO and be able to reduce inflammation and oxidative stress in epithelial intestinal cells. In vivo studies are necessary to validate the in vitro results and to investigate the rate of fermented rape seed meal inclusion in the diet.

## 5. Materials and Methods

### 5.1. Fermented Rapeseed Meal Extract and ZnO Solutions

Rapeseed meal provided by EXPUR S.A., Slobozia, Romania was fermented with *Saccharomyces boulardi* (CNCM I– 745, Gentilly, France) using the method described by [19]. Briefly, rapeseed meal was homogenized with water and yeast for 15 min and fermented for 24 h in glass jars. The fermented rapeseed meal was then washed with distilled water for 5 min and decanted. The excess water was collected to remove the soluble antinutrients (glucosinolates and derivatives) and the wet meal was dried at 70 °C. One gram of fermented rapeseed meal was mixed with 7 mL (ratio of sample: solvent 1:7 *w*/*v*) of 80% acetone and shaken continuously for 24 h. Samples were concentrated to remove the acetone and aqueous extract was diluted and used in co-cultured cells.

ZnO was dissolved in 5% acetic acid, resulting a 400 mM stock solution. Following this step, the 400 mM ZnO solution was diluted in the co-culture medium to a final concentration of 50 µM.

### 5.2. Cell Culture and Treatments

Caco-2 cells (American Type Culture Collection, Manassas, Virginia, USA) were cultured in Modified Eagle Medium (MEM) supplemented with 10% fetal bovine serum, 1% antibiotic–antimycotic solution and L-glutamine (all from Sigma Aldrich, St. Louis, MO, USA). HT29-MTX cells (ECACC, European Collection of Authenticated Cell Cultures, Salisbury, UK) were cultured in RPMI-1640 medium containing 10% fetal bovine serum, 1% (*v*/*v*) nonessential amino acids, 2 mM L-glutamine, and 1% antibiotic–antimycotic solution. For co-culture experiments, the cells were seeded in a 9:1 (Caco-2:HT29-MTX) ratio [51] and cultured until confluence in MEM medium containing 10% fetal bovine serum, 1% (*v*/*v*) nonessential amino acids, 2 mM L-glutamine, and 1% antibiotic–antimycotic solution at 37 °C in a 5% CO_2_ humidified atmosphere, with the culture medium being changed every three days. The cells were cultured at a concentration of 0.1 × 10^6^ cells/mL in 24-well culture plates (Nunc). Cells were microscopically visualized daily and when confluence was achieved, the cells were stimulated with a cocktail of cytokines (IFN-γ: 50 ng/mL, TNF-α: 50 ng/mL, IL-1β: 25 ng/mL). An MTT test with different dilutions (1/10, 1/25, 1/50, and 1/100) of the FRSM extract was performed to choose the appropriate dilution to be used in the in vitro experiments. Cell viability was over 90% for all dilutions tested. The concentration of ZnO was chosen according to [4]. Cellular co-cultures were then treated with FRSM extract (1/50 dilution) and ZnO (50 µM). After 4 h incubation, cells were treated with 1 µg/mL LPS and incubated for another 24 h. Six treatments resulted as follows:(1)Untreated cells (control),(2)LPS = cells treated with 1 µg/mL LPS after 4 h of incubation until 24 h,(3)ZnO = cells treated with ZnO (50 µM) 24 h,(4)ZnO + LPS = cells pretreated with ZnO for 4 h and then challenge with LPS for another 24 h,(5)FRSM = cells treated with FRSM (1/50) 24 h,(6)FRSM + LPS = cells pretreated with FRSM for 4 h and then challenge with LPS for another 24 h.

For flow cytometry analyses, the cells were detached with ethylenediaminetetraacetic acid (EDTA)-trypsin, washed with sterile PBS, and used immediately. All tests were carried out in three individual experiments with two replicates per treatment.

### 5.3. Detection of Apoptosis

Flow cytometry and Muse^®^ Annexin V & Dead Cell Kit (Luminex Corporation, Austin, TX, USA) were used for the quantitative analysis of live, early, late apoptosis, and cell death of Caco-2/HT-29 MTX co-cultured cells treated with ZnO and FRSM extract. After the cell staining, the live, early apoptotic, late apoptotic, total apoptotic, and dead cells were acquired using a Guava^®^ Muse^®^ Cell Analyzer. The experiments were performed in triplicate. After the dot plot analysis, the results were presented as percentages of live, early apoptotic, late apoptotic, total apoptotic, and dead cells of total cells.

### 5.4. Detection of Oxidative Stress Markers

#### 5.4.1. ROS Production

Reactive Oxygen Species (ROS)-positive (ROS+) and -negative cells (ROS−) in Caco-2/HT-29 MTX co-cultured cells were detected by flow cytometry. Cells were stained using Muse^®^ Oxidative Stress Kit (Luminex, Austin, TX, USA). The acquisition was performed using Guava^®^ Muse^®^ Cell Analyzer and the experiments were performed in triplicate. The results were presented as percentages of ROS+ and ROS- cells.

#### 5.4.2. Lipid Peroxidation (Thiobarbituric-Acid-Reactive Substances—TBARS) Analysis

The concentration of thiobarbituric-acid-reactive substances (TBARS) was measured in cell culture supernatants using the method described by [52]. Results were expressed as nmol/mL.

#### 5.4.3. Protein Oxidation (Protein Carbonyl) Analysis

The protein carbonyl content in co-cultured cells was evaluated using the Protein carbonyl colorimetric kit (Cayman Chemical, Ann Arbor, MI, USA), based on the spectrophotometric detection of the protein hydrazone as product of the reaction between 2.4-dinitrophenyl hydrazine with protein carbonyls. The protein concentrations of samples were determined using the Pierce method (Pierce BCA Protein Assay Kit, ThermoFisher Scientific, Rockford, IL, USA). The absorbance of samples was determined at a test wavelength of 370 nm using an ELISA microplate reader (Varioskan™ LUX, ThermoFisher Scientific, Rockford, IL, USA). All experiments were performed in three independent replicates. The results were expressed as carbonyl content (nmol/mg) = (carbonyl nmol/mL)/(protein mg/mL).

### 5.5. Detection of Inflammatory and Signaling Pathway Markers

#### Determination of Protein Expression (Protein Array)

Human Cytokine Antibody Array (ab133997, Abcam, Cambridge, UK) was used for the simultaneous detection of 19 cytokines and 12 chemokines in cellular co-culture supernatants, according to the manufacturer’s recommendations. Briefly, the array membranes were incubated in blocking buffer for 30 min at room temperature. Next, 1 mL of cell cultured supernatant was added to the membranes and were incubated overnight at 4 °C on a rocking platform shaker. The membranes were further washed and incubated in Biotin-Conjugated Anti-Cytokines over night at room temperature. HRP-conjugated streptavidin was added to the arrays and incubated for 2 hours at room temperature. The membrane arrays were developed with Chemiluminescence Detection reagents and images were captured on MicroChemi System (DNR Bio-Imaging Systems). The signal density (pixel density) of each spot on the array was quantified using Image J software (https://imagej.nih.gov/ij/ (accessed on 15 June 2022)), and expressed as Arbitrary Units (AU).

The levels of signaling phosphoproteins were detected using the Proteome Profiler Human Phospho-Kinase Array Kit (ARY003C, R&D Systems, Minneapolis, MN, USA) according to the manufacturer’s instructions. CaCO-2/HT-29MTX co-cultured cells were lysed and 300 μg of protein lysates were used for the Phospho-Kinase arrays. The signals were detected using MicroChemi System (DNR Bio-Imaging Systems, Israel) and were quantified using Image J software (https://imagej.nih.gov/ij/ (accessed on 15 June 2022)) from three independent replicates.

### 5.6. Gelatine Zymography

The gelatinolytic MMPs (MMP-2 and MMP-9) activities in cell culture supernatants were detected by gelatine zymography. The Caco-2/HT-29 MTX cells were co-cultured in serum-free medium and treated as described above. At the end of incubation, the supernatants were collected and analyzed for total protein content using a commercial kit (Pierce BCA Protein Assay Kit ThermoFisher Scientific, Rockford, IL, USA). First, 30 µg of protein was diluted in Zymogram sample buffer (Bio-Rad Laboratories, Inc., Hercules, CA, USA) and subjected to 10% SDS-PAGE electrophoresis in the presence of 0.1% gelatine. After electrophoresis, the proteins were renatured by incubation of gels in Zymogram renaturation buffer (Bio-Rad Laboratories, Inc., Hercules, CA, USA) for 30 min. The enzyme activities in Zymogram gels were initiated by incubation of the gels in Zymogram development buffer (BioRad), at 37 °C for 18 h. The Zymograms were stained by incubation with Coomassie Brilliant Blue R250 for one hour, and in the next step were incubated with destaining solution (10% methanol/5% acetic acid, *v*/*v*) until the clear bands on a blue background were clearly seen. The gels were scanned and the MMPs’ activities were quantified using ImageLab software (BioRad), and all the obtained results were expressed as arbitrary units (AU). Fetal bovine serum (FBS) was used as positive control for both MMP-2 and MMP-9 activity.

### 5.7. Statistical Analysis

The data are presented as means ± standard error of the mean. The differences between treatments for all parameters analyzed were compared using StatView software 6.0 (SAS Institute, Cary, NC, USA) with one-way ANOVA and Student’s *t*-test followed by Fisher’s procedure of the least square difference. Values of *p* < 0.05 were considered significant.

## Figures and Tables

**Figure 1 ijms-23-11640-f001:**
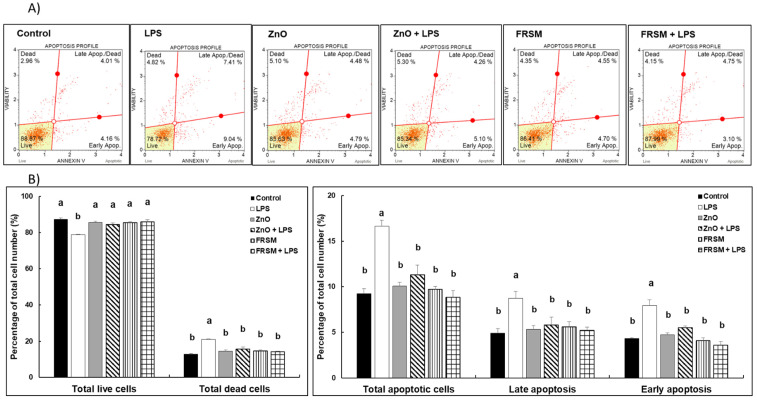
**Effect of ZnO and FRSM extract on apoptosis in co-cultured Caco-2/HT29MTX challenged with LPS.** Co-cultured cells were incubated with FRSM extract (1/50 dilution) and ZnO (50 µM) either alone or followed by challenge after 4 h with 1 µg/mL LPS and further incubation for another 24 h. Apoptosis profiling was performed using Muse Annexin & Dead Cell kit. Results are presented as percentages of live, early apoptotic, late apoptotic, total apoptotic, and dead cells of total cells. (**A**) Representative dot plots sorted by flow cytometry and (**B**) quantification of apoptotic cells corresponding to each treatment. ^a,b^ Histograms with different superscript letters were significantly different (*p* < 0.050). Control = untreated control cells; LPS = cells treated with 1 µg/mL LPS; ZnO = cells treated with ZnO (50 µM); ZnO + LPS = cells treated with ZnO 4 h + LPS for another 24 h; FRSM = cells treated with FRSM (1/50); FRSM + LPS = cells treated with FRSM 4 h + LPS for another 24 h.

**Figure 2 ijms-23-11640-f002:**
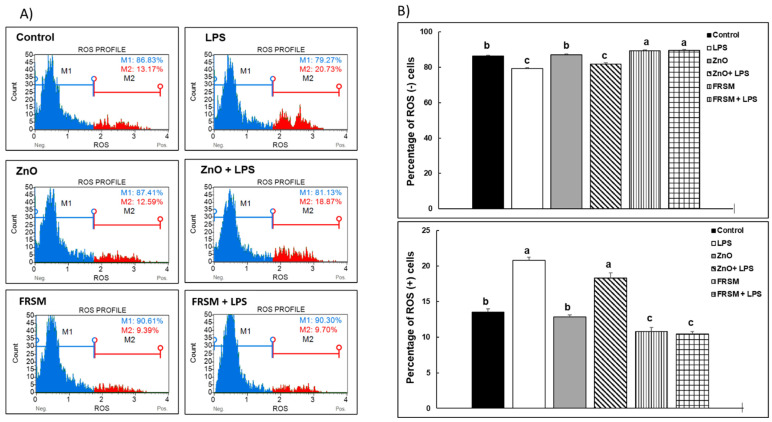
**Effect of ZnO and FRSM extract on ROS production in co-cultured Caco-2/HT29MTX challenged with LPS.** Co-cultured cells were incubated with FRSM extract (1/50 dilution) and ZnO (50 µM) either alone or followed by challenge after 4 h with 1 µg/mL LPS and further incubation for another 24 h. ROS production was analyzed using the Muse Annexin & Dead Cell kit. Three replicates/treatment were analyzed, and median values of ROS (+) and ROS (−) cell percentages were expressed. (**A**) Representative flow cytometry assay showing staining distribution among the different treatments and (**B**) quantification of ROS (+) and ROS (−) cells. ^a,b,c^ Histograms for each treatment with different superscript letters were significantly different (*p* < 0.050). Control = untreated control cells; LPS = cells treated with 1 µg/mL LPS; ZnO = cells treated with ZnO (50 µM); ZnO + LPS = cells treated with ZnO 4 h + LPS for another 24 h; FRSM = cells treated with FRSM (1/50); FRSM + LPS = cells treated with FRSM 4 h + LPS for another 24 h.

**Figure 3 ijms-23-11640-f003:**
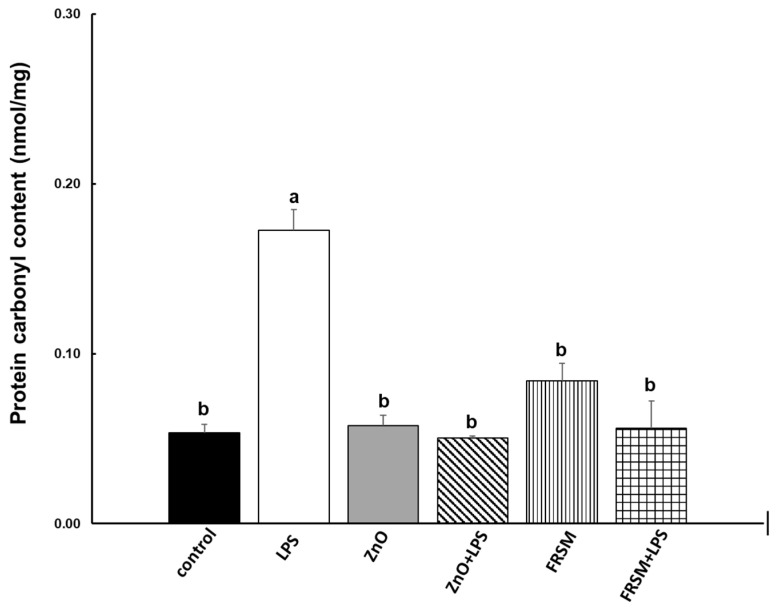
**Effect of ZnO and FRSM extract on protein oxidation in co-cultured Caco-2/HT29MTX challenged with LPS (protein carbonyl).** Co-cultured cells were incubated with FRSM extract (1/50 dilution) and ZnO (50 µM) either alone or followed by challenge after 4 h with 1 µg/mL LPS and further incubation for another 24 h. Results are expressed as means ± standard error (SEM). ^a,b^ Histograms for each group with different superscript letters were significantly different (*p* < 0.050). Control = untreated control cells; LPS = cells treated with 1 µg/mL LPS; ZnO = cells treated with ZnO (50 µM); ZnO + LPS = cells treated with ZnO 4 h + LPS for another 24 h; FRSM = cells treated with FRSM (1/50); FRSM + LPS = cells treated with FRSM 4 h + LPS for another 24 h.

**Figure 4 ijms-23-11640-f004:**
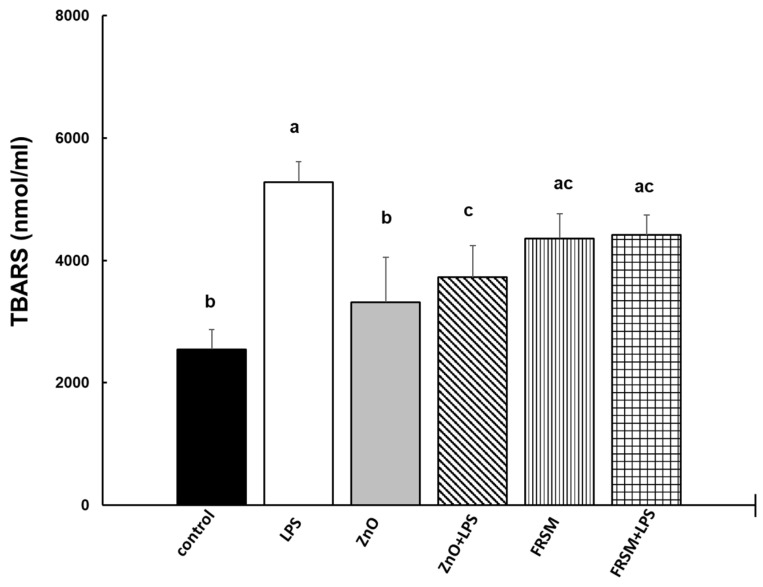
**Effect of ZnO and FRSM extract on lipid oxidation in co-cultured Caco-2/HT29MTX challenged with LPS (TBARS).** Co-cultured cells were incubated with FRSM extract (1/50 dilution) and ZnO (50 µM) either alone or followed by challenge after 4 h with 1 µg/mL LPS and further incubation for another 24 h. Results are expressed as means ± standard error (SEM). ^a,b,c^ Histograms for each group with different superscript letters were significantly different (*p* < 0.050). Control = untreated control cells; LPS = cells treated with 1 µg/mL LPS; ZnO = cells treated with ZnO (50 µM); ZnO + LPS = cells treated with ZnO 4 h + LPS for another 24 h; FRSM = cells treated with FRSM (1/50); FRSM + LPS = cells treated with FRSM 4 h + LPS for another 24 h.

**Figure 5 ijms-23-11640-f005:**
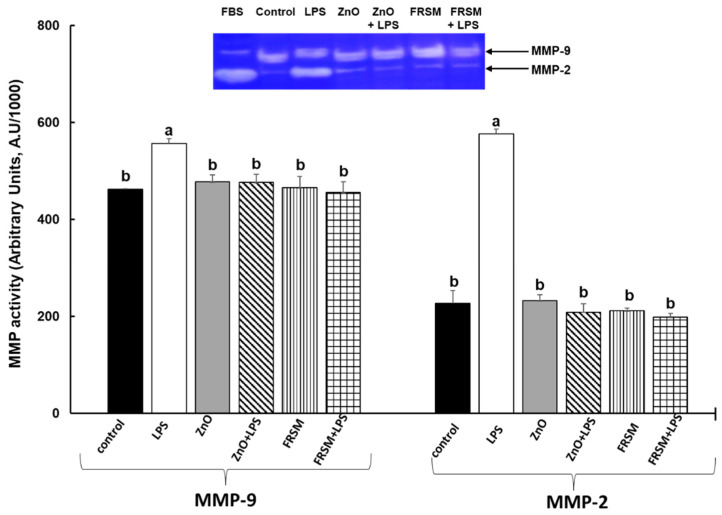
**Effect of ZnO and FRSM extract on metalloproteinase activity in co-cultured Caco-2/HT29MTX challenged with LPS.** Co-cultured cells were incubated with FRSM extract (1/50 dilution) and ZnO (50 µM) alone or followed by challenge after 4 h with 1 µg/mL LPS and further incubation for another 24 h. Results are expressed as means ± standard error (SEM). ^a,b^ Histograms for each group with different superscript letters were significantly different (*p* < 0.050). Control = untreated control cells; LPS = cells treated with 1 µg/mL LPS; ZnO = cells treated with ZnO (50 µM); ZnO + LPS = cells treated with ZnO 4 h + LPS for another 24 h; FRSM = cells treated with FRSM (1/50); FRSM + LPS = cells treated with FRSM 4 h + LPS for another 24 h.

**Table 1 ijms-23-11640-t001:** Overview evaluation of the effect of ZnO and FRSM treatments on global protein expression after LPS action.

Functional Classification		Experimental Treatments					
	Expression Level	Control (%)	LPS (%)	ZnO (%)	FRSM (%)	ZnO + LPS (%)	FRSM + LPS (%)
**Cytokines** (19 proteins)	Overexpressed (%)	0.00 ^a^	17.91 ^b^ ± 2.9	0.00 ^a^	0.00 ^a^	0.00 ^a^	1.49 ^c^ ± 0.9
	Suppressed (%)	0.00 ^a^	0.00 ^a^	7.46 ^c^	5.97 ^d^	11.94 ^b^	5.97 ^d^
	No effect (%)	0.00 ^a^	10.46 ^d^ ± 5.1	20.89 ^b^ ± 4.2	22.39 ^b^ ± 3.9	16.42 ^c^ ± 3.7	20.90 ^b^ ± 1.9
**Chemokines** (12 proteins)	Overexpressed (%)	0.00 ^a^	11.94 ^b^ ± 1.2	0.00 ^a^	0.00 ^a^	0.00 ^a^	0.0 ^a^
	Suppressed (%)	0.00 ^a^	0.00 ^a^	1.49 ^c^ ± 0.4	1.49 ^c^ ± 0.7	4.48 ^b^ ± 1.0	1.49 ^c^ ± 0.2
	No effect (%)	0.00 ^a^	5.97 ^c^ ± 1.1	16.43 ^b^ ± 2.7	16.42 ^b^ ± 1.9	13.43 ^b^ ± 1.5	16.41 ^b^ ± 1.3
**Signaling molecules** (36 proteins)	Overexpressed (%)	0.00 ^a^	35.82 ^b^ ± 3.9	19.40 ^c^ ± 2.0	11.94 ^c^ ± 1.7	10.45 ^c^ ± 1.6	8.96 ^c^ ± 1.3
	Suppressed (%)	0.00 ^a^	1.49 ^f^ ± 0.8	4.48 ^d^ ± 0.8	8.95 ^c^ ± 1.2	2.98 ^e^ ± 0.3	13.44 ^b^ ± 2.2
	No effect (%)	0.00 ^a^	16.41 ^d^ ± 2.1	29.85 ^c^ ± 4.3	32.84 ^c^ ± 3.7	40.30 ^b^ ± 2.8	31.34 ^c^ ± 3.3

Co-cultured cells were incubated with FRSM extract (1/50 dilution) and ZnO (50 µM) either alone or followed by challenge after 4 h with 1 µg/mL LPS and further incubation for another 24 h. Results are expressed as means ± standard deviation (SD) and standard error (SEM). Values within a row with different superscript letters were significantly different (*p* < 0.050). Control = untreated control cells; LPS = cells treated with 1 µg/mL LPS; ZnO = cells treated with ZnO (50 µM); ZnO + LPS = cells treated with ZnO 4 h + LPS for another 24 h; FRSM = cells treated with FRSM (1/50); FRSM + LPS = cells treated with FRSM 4 h + LPS for another 24 h.

**Table 2 ijms-23-11640-t002:** The effect of ZnO and FRSM extract on protein expressions of pro-, anti-inflammatory, and inflammation-regulation cytokines in co-cultured Caco-2/HT29-MTX.

		Experimental Treatments						
Cytokines Involvement	Protein Name	Control (%)	LPS (%)	ZnO (%)	FRSM (%)	ZnO + LPS (%)	FRSM + LPS (%)	SEM
Inflammatory response	IL-1α	100 ^a^	142.9 ^b^ ± 33.2	63.4 ^c^ ± 18.5	74.0 ^abc^ ± 12.3	62.6 ^c^ ± 9.3	126.3 ^ab^ ± 31.2	10.4
	IL-1β	100 ^ab^	141.8 ^b^ ± 18.5	101.5 ^ab^ ± 25.6	91.8 ^a^ ± 21.0	82.2 ^a^ ± 20.0	82.3 ^a^ ± 10.2	9.6
	IL-2	100 ^a^	112.1 ^a^ ± 21.8	49.6 ^b^ ± 10.9	57.8 ^b^ ± 9.2	55.9 ^b^ ± 17.8	59.1 ^b^ ± 10.5	2.7
	IL-6	100 ^ab^	126.7 ^a^ ± 20.7	69.1 ^ab^ ± 10.5	111.8 ^ab^ ± 48.4	61.0 ^b^ ± 12.4	102.7 ^a^ ± 28.5	9.5
	IL-12 p40/p70	100 ^a^	138.1 ^b^ ± 20.3	102.6 ^ab^ ± 40.3	104.7 ^ab^ ± 24.5	85.4 ^a^ ± 27.7	104.6 ^ab^ ± 15.8	6.3
	IL-15	100 ^a^	141.6 ^b^ ± 17.0	99.6 ^ab^ ± 11.5	104.5 ^ab^ ± 21.5	78.4 ^ab^ ± 16.2	111.7 ^ab^ ± 19.8	8.2
	TNF-α	100 ^a^	165.0 ^b^ ± 20.9	114.9 ^ab^ ± 16.2	102.9 ^a^ ± 20.2	129.3 ^ab^ ± 11.4	114.9 ^ab^ ± 16.3	4.1
	TNF-β	100 ^a^	109.7 ^a^ ± 7.9	65.3 ^b^ ± 17.7	59.3 ^bc^ ± 14.6	40.1 ^c^ ± 6.41	47.9 ^bc^ ± 6.4	5.8
	IFN-γ	100 ^a^	141.7 ^b^ ± 15.9	116.3 ^ab^ ± 18.7	111.8 ^ab^ ± 5.7	107.1 ^a^ ± 14.5	106.1 ^a^ ± 8.2	1.8
	MCSF	100 ^a^	154.3 ^b^ ± 25.9	108.9 ^a^ ± 28.9	101.5 ^a^ ± 26.6	101.8 ^a^ ± 17.8	108.3 ^a^ ± 24.4	5.3
	GCSF	100 ^a^	193.1 ^b^ ± 13.9	79.2 ^a^ ± 19.7	88.1 ^a^ ± 18.1	89.3 ^a^ ± 15.4	152.5 ^b^ ± 12.3	2.7
	OncostainM	100 ^a^	148.3 ^b^ ± 19.4	95.5 ^a^ ± 15.6	113.3 ^ab^ ± 23.4	79.8 ^a^ ± 11.9	108.68 ^a^ ± 6.2	7.6
Anti-inflammatory response	IL-4	100 ^ab^	101.4 ^a^ ± 13.9	79.5 ^b^ ± 14.7	83.5 ^ab^ ± 24.9	59.4 ^c^ ± 12.7	92.5 ^ab^ ± 8.3	9.6
	IL-10	100 ^a^	95.1 ^a^ ± 16.7	85.2 ^a^ ± 18.9	75.1 ^ab^ ± 19.4	63.0 ^b^ ± 14.0	95.1 ^a^ ± 16.7	7.6
	IL-13	100 ^a^	127.4 ^a^ ± 23.4	52.2 ^b^ ± 12.8	77.4 ^ab^ ± 12.9	65.6 ^b^ ± 11.8	96.0 ^a^ ± 15.6	10.3
Regulation of inflammatory response	IL-3	100 ^a^	161.9 ^b^ ± 34.1	88.0 ^a^ ± 21.03	85.9 ^a^ ± 16.1	81.1 ^a^ ± 15.1	91.1 ^a^ ± 17.5	3.9
	IL-7	100 ^a^	204.3 ^b^ ± 17.2	59.4 ^c^ ± 18.2	63.8 ^c^ ± 3.6	55.6 ^c^ ± 8.6	59.2 ^c^ ± 4.6	7.6
	IL-5	100 ^a^	189.5 ^b^ ± 27.3	58.4 ^c^ ± 9.4	73.2 ^c^ ± 10.9	47.0 ^c^ ± 6.9	56.4 ^c^ ± 2.7	4.9
	GM-CSF	100 ^ab^	139.0 ^b^ ± 29.3	76.8 ^ab^ ± 18.3	84.2 ^b^ ± 17.4	74.6 ^b^ ± 7.6	112.6 ^ab^ ± 27.5	10.1

Co-cultured cells were incubated with FRSM extract (1/50 dilution) and ZnO (50 µM) alone or followed by challenge after 4 h with 1 µg/mL LPS and further incubation for another 24 h. Results are expressed as means ± standard deviation (SD) and standard error (SEM). Values within a row with different superscript letters were significantly different (*p* < 0.050). IL = interleukin; TNF-α = tumor necrosis factor alpha; MCSF = macrophage colony-stimulating factor; GCSF = Granulocyte colony-stimulating factor; GM-CSF = Granulocyte-macrophage colony-stimulating factor. Control = untreated control cells; LPS = cells treated with 1 µg/mL LPS; ZnO = cells treated with ZnO (50 µM); ZnO + LPS = cells treated with ZnO 4 h + LPS for another 24 h; FRSM = cells treated with FRSM (1/50); FRSM + LPS = cells treated with FRSM 4 h + LPS for another 24 h.

**Table 3 ijms-23-11640-t003:** The effect of ZnO and FRSM extract on protein expressions of chemokines in co-cultured Caco-2/HT29-MTX.

	Experimental Treatments							
Chemokine Type	Protein Name	Control (%)	LPS (%)	ZnO (%)	FRSM (%)	ZnO + LPS (%)	FRSM + LPS (%)	SEM
CXC	IL-8	100 ^a^	161.3 ^b^ ± 28.3	108.9 ^a^ ± 35.2	98.4 ^a^ ± 26.1	111.7 ^a^ ± 24.6	102.6 ^a^ ± 25.5	9.9
	MIG	100 ^a^	132.2 ^b^ ± 36.9	110.7 ^a^ ± 32.0	102.6 ^a^ ± 28.2	114.4 ^a^ ± 37.3	115.8 ^a^ ± 28.8	9.8
	ENA-78	100 ^ab^	128.6 ^a^ ± 10.1	98.3 ^ab^ ± 18.6	79.0 ^b^ ± 19.2	93.4 ^b^ ± 12.2	72.2 ^b^ ± 14.2	6.6
	GRO-α	100 ^a^	134.3 ^b^ ± 35.9	104.7 ^a^ ± 29.4	95.5 ^a^ ± 15.9	101.7 ^a^ ± 21.3	95.3 ^a^ ± 9.5	8.3
CCL	MCP-1	100 ^a^	152.8 ^b^ ± 18.9	107.1 ^a^ ± 11.1	103.1 ^a^ ± 3.3	119.0 ^ab^ ± 9.1	108.8 ^a^ ± 17.0	6.0
	MCP-2	100 ^a^	125.2 ^a^ ± 8.8	64.4 ^b^ ± 15.3	79.1 ^ab^ ± 20.2	57.7 ^b^ ± 13.2	74.8 ^ab^ ± 23.5	7.8
	MCP-3	100 ^a^	133.8 ^b^ ± 19.0	74.1 ^ac^ ± 20.2	72.8 ^ac^ ± 30.3	55.7 ^c^ ± 15.2	75.6 ^ac^ ± 27.1	10.3
	MIP-1δ	100 ^a^	130.7 ^b^ ± 20.6	98.3 ^a^ ± 11.7	94.9 ^a^ ± 5.7	117.6 ^ab^ ± 5.4	89.2 ^a^ ± 7.8	5.5
	RANTES	100 ^a^	147.1 ^b^ ± 18.4	113.3 ^a^ ± 21.6	93.5 ^a^ ± 9.2	88.2 ^a^ ± 19.7	77.5 ^a^ ± 26.5	8.4
	I-309	100 ^ab^	120.2 ^a^ ± 25.5	75.4 ^b^ ± 27.3	83.5 ^b^ ± 20.2	49.3 ^c^ ± 4.4	85.2 ^b^ ± 18.9	10.5
	MDC	100 ^a^	134.3 ^b^ ± 15.6	94.9 ^a^ ± 14.9	64.5 ^c^ ± 23.1	73.1 ^a^ ± 10.5	64.0 ^c^ ± 17.7	8.4
	TARC	100 ^a^	121.5 ^a^ ± 31.7	81.4 ^ab^ ± 14.1	85.8 ^ab^ ± 18.1	58.5 ^b^ ± 10.7	79.7 ^ab^ ± 11.9	8.4

Co-cultured cells were incubated with FRSM extract (1/50 dilution) and ZnO (50 µM) alone or followed by challenge after 4 h with 1 µg/mL LPS and further incubation for another 24 h. Results are expressed as means ± standard deviation (SD) and standard error (SEM). Values within a row with different superscript letters were significantly different (*p* < 0.050). Control = untreated control cells; LPS = cells treated with 1 µg/mL LPS; ZnO = cells treated with ZnO (50 µM); ZnO + LPS = cells treated with ZnO 4 h + LPS for another 24 h; FRSM = cells treated with FRSM (1/50); FRSM + LPS = cells treated with FRSM 4 h + LPS for another 24 h.

**Table 4 ijms-23-11640-t004:** (**A**) The effect of ZnO and FRSM extract on signaling molecule protein expression in co-cultured Caco-2/HT 29-MTX. (**B**) The effect of ZnO and FRSM extract on signaling molecule protein expression in co-cultured Caco-2/HT29-MTX.

**(A)**								
		**Experimental Treatments**						
**Pathway** **Involvement**	**Protein Name**	**Control (%)**	**LPS (%)**	**ZnO (%)**	**FRSM (%)**	**ZnO + LPS (%)**	**FRSM + LPS (%)**	**SEM**
Inflammation/ oxidative stress/apoptosis	MAPK-p38-α	100 ^a^	226.2 ^b^ ± 29.1	105.5 ^a^ ± 16.8	139.1 ^a^ ± 20.6	122.3 ^a^ ± 25.0	167.9 ^b^ ± 23.7	19.6
	JNK1/2/3	100 ^a^	158.4 ^b^ ± 31.7	93.4 ^a^ ± 14.8	107.5 ^a^ ± 10.8	106.6 ^a^ ± 21.2	99.6 ^a^ ± 11.2	13.8
	MSK1/2	100 ^ac^	130.5 ^b^ ± 19.4	100.9 ^a^ ± 14.1	103.4 ^a^ ± 16.9	83.62 ^a^ ± 25.1	120.8 ^a^ ± 19.6	16.2
	GSK-3β	100 ^a^	78.6 ^b^ ± 5.81	82.2 ^ab^ ± 14.9	77.3 ^ab^ ± 20.6	74.2 ^ab^ ± 17.2	65.6 ^b^ ± 9.8	4.7
	HSP27	100 ^a^	145.5 ^b^ ± 25.3	96.3 ^a^ ± 11.7	116.3 ^a^ ± 35.3	82.67 ^a^ ± 10.9	116.1 ^a^ ± 22.3	8.6
	Src	100 ^a^	147.5 ^b^ ± 25.9	116.4 ^a^ ± 22.7	122.7 ^a^ ± 30.0	106.2 ^a^ ± 22.7	123.3 ^a^ ± 18.7	8.9
	Lck	100 ^a^	67.6 ^b^ ± 11.1	75.7 ^ab^ ± 14.5	112.0 ^a^ ± 14.9	97.9 ^a^ ± 15.3	84.3 ^ab^ ± 16.7	7.2
	CREB	100 ^a^	193.8 ^b^ ± 34.7	111.4 ^a^ ± 13.7	103.0 ^a^ ± 11..2	102.5 ^b^ ± 13.0	103.9 ^b^ ± 26.8	18.4
	c-Jun	100 ^a^	275.7 ^b^ ± 36.4	85.4 ^ac^ ± 9.7	93.4 ^a^ ± 12.8	77.8 ^c^ ± 11.6	75.6 ^d^ ± 16.1	16.5
Transcription factors	STAT1	100 ^a^	112.9 ^a^ ± 24.7	154.6 ^b^ ± 15.7	85.3 ^a^ ± 11.5	93.4 ^a^ ± 8.9	94.8 ^a^ ± 18.6	17.1
	STAT2	100 ^ac^	203.8 ^b^ ± 33.6	115.6 ^a^ ± 21.9	86.3 ^c^ ± 10.0	126.1 ^a^ ± 21.7	127.0 ^a^ ± 31.7	20.7
	STAT3	100 ^a^	241.3 ^b^ ± 23.9	202.6 ^b^ ± 20.1	127.0 ^a^ ± 13.6	115.2 ^a^ ± 12.4	106.0 ^a^ ± 22.1	13
	STAT5 α/β	100 ^a^	110.7 ^a^ ± 30.6	99.9 ^a^ ± 31.7	83.9 ^a^ ± 14.0	81.5 ^a^ ± 10.5	97.5 ^a^ ± 14.5	19.8
	STAT6	100 ^a^	131.3 ^b^ ± 28.2	127.7 ^b^ ± 26.1	102.9 ^a^ ± 33.7	100.4 ^a^ ± 16.0	67.6 ^c^ ± 11.0	13.5
	beta-catenin	100 ^a^	118.3 ^a^ ± 11.7	116.3 ^a^ ± 10.3	109.1 ^a^ ± 24.2	84.3 ^a^ ± 15.2	86.4 ^a^ ± 14.6	17.5
Cellular migration/ chemotaxis	PYK2	100 ^a^	144.7 ^b^ ± 31.1	145.6 ^b^ ± 10.1	89.3 ^ac^ ± 10.9	93.9 ^a^ ± 10.3	71.9 ^c^ ± 13.9	16.9
	PDGF Rβ	100 ^a^	89.4 ^a^ ± 12.6	103.1 ^a^ ± 20.6	102.6 ^a^ ± 15.6	83.71 ^a^ ± 22.0	104.6 ^a^ ± 11.0	4.71
	PLC-γ1	100 ^a^	150.5 ^b^ ± 21.9	105.3 ^a^ ± 25.5	103.9 ^a^ ± 27.7	79.5 ^a^ ± 19.4	108.7 ^a^ ± 14.4	9
	Fgr	100 ^a^	96.5 ^a^ ± 10.7	102.2 ^a^ ± 24.6	91.3 ^a^ ± 13.7	80.2 ^a^ ± 17.3	96.6 ^a^ ± 14.8	3.5
**(B)**								
**Pathway Involvement**	**Protein Name**	**Control (%)**	**LPS (%)**	**ZnO (%)**	**FRSM (%)**	**ZnO + LPS (%)**	**FRSM + LPS (%)**	**SEM**
Cell proliferation/cell cycle/ differentiation	RSK1/2	100 ^a^	142.0 ^b^ ± 21.9	136.4 ^b^ ± 16.6	67.5 ^c^ ± 10.3	73.8 ^c^ ± 12.1	56.2 ^d^ ± 13.2	18.2
	RSK1/2/3	100 ^a^	147.9 ^b^ ± 27.1	139.1 ^b^ ± 16.3	59.0 ^c^ ± 7.5	117.9 ^a^ ± 26.0	66.1 ^c^ ± 14.4	17.7
	ERK1/2	100 ^a^	114.9 ^ab^ ± 28.9	90.0 ^ab^ ± 21.9	80.3 ^b^ ± 17.6	53.8 ^c^ ± 9.3	71.8 ^b^ ± 19.8	21.6
	Chk-2	100 ^a^	174.3 ^b^ ± 35.4	129.3 ^a^ ± 21.9	139.0 ^c^ ± 25.9	93.3 ^a^ ± 10.9	83.0 ^c^ ± 12.1	20.8
	p53 (S15)	100 ^a^	156.4 ^b^ ± 33.7	139.4 ^c^ ± 16.5	83.9 ^a^ ± 15.9	84.6 ^a^ ± 17.5	83.3 ^a^ ± 18.8	16.5
	p53 (S392)	100 ^a^	169.4 ^b^ ± 36.5	143.8 ^bc^ ± 16.1	124.7 ^ac^ ± 30.4	150.3 ^b^ ± 27.1	135.2 ^c^ ± 23.6	12.1
	p53 (S46)	100 ^a^	99.7 ^a^ ± 7.2	141.6 ^b^ ± 28.3	54.8 ^d^ ± 10.4	110.8 ^a^ ± 22.7	66.9 ^c^ ± 16.4	15.6
	EGF R	100 ^a^	90.4 ^a^ ± 27.8	64.3 ^b^ ± 12.5	98.8 ^a^ ± 25.1	84.3 ^a^ ± 15.9	83.2 ^a^ ± 12.5	13.2
	p70S6k (T389)	100 ^a^	174.3 ^b^ ± 20.5	154.9 ^b^ ± 22.0	94.2 ^a^ ± 29.4	99.9 ^a^ ± 20.7	70.1 ^c^ ± 19.5	18.2
	p70S6k (T421/S42)	100 ^a^	146.8 ^b^ ± 23.9	134.1 ^b^ ± 30.0	77.6 ^c^ ± 16.8	76.1 ^c^ ± 10.6	70.3 ^c^ ± 10.1	15.2
	Akt1/2/3 (S473)	100 ^a^	171.7 ^b^ ± 9.2	114.0 ^a^ ± 29.9	69.7 ^c^ ± 20.6	86.9 ^a^ ± 20.7	73.0 ^c^ ± 13.1	15.7
	Akt1/2/3 (T308)	100 ^ac^	171.6 ^b^ ± 16.0	117.1 ^a^ ± 25.9	63.0 ^d^ ± 9.5	85.9 ^ac^ ± 14.7	79.0 ^c^ ± 14.3	16.4
	PRAS40	100 ^a^	121.8 ^b^ ± 26.3	135.9 ^b^ ± 33.0	87.0 ^ac^ ± 26.7	105.7 ^a^ ± 22.5	66.2 ^d^ ± 11.4	15.1
Other kinases	eNOS	100 ^a^	214.4 ^b^ ± 38.6	152.3 ^c^ ± 30.2	116.9 ^a^ ± 18.0	99.3 ^a^ ± 19.3	99.4 ^a^ ± 11.9	15.6
	Yes	100 ^a^	133.4 ^c^ ± 17.9	97.9 ^a^ ± 11.4	199.7 ^b^ ± 31.7	89.4 ^a^ ± 16.0	98.2 ^a^ ± 18.9	10.3
	GSK3α/β	100 ^a^	89.5 ^b^ ± 12.5	86.8 ^b^ ± 24.1	101.2 ^a^ ± 11.4	90.1 ^ab^ ± 18.0	91.5 ^c^ ± 11.5	8.9
	Lyn	100 ^a^	107.3 ^a^ ± 21.9	109.8 ^a^ ± 20.9	78.0 ^c^ ± 20.2	118.6 ^a^ ± 27.7	65.4 ^d^ ± 7.6	6.1

(A) Co-cultured cells were incubated with FRSM extract (1/50 dilution) and ZnO (50 µM) alone or followed by challenge after 4 h with 1 µg/mL LPS and further incubation for another 24 h. Results are expressed as means ± standard deviation (SD) and standard error (SEM). Values within a row with unlike superscript letters were significantly different (*p* < 0.050). Control = untreated control cells; LPS = cells treated with 1 µg/mL LPS; ZnO = cells treated with ZnO (50 µM); ZnO + LPS = cells treated with ZnO 4 h + LPS for another 24 h; FRSM = cells treated with FRSM (1/50); FRSM + LPS = cells treated with FRSM 4 h + LPS for another 24 h. p38-MAPK = p38 mitogen-activated protein kinases; JNK = c-Jun-N-terminale Kinase; MSK = Mitogen and stress activated protein kinase; GSK-β = Glycogen synthase kinase-3 beta; HSP27 = Heat shock protein 27; Src = Proto-oncogene tyrosine-protein kinase Src; LcK = lymphocyte-specific protein tyrosine kinase; STAT1-6 = signal transducer and activator of transcription (STAT) protein family; PYK2 = Protein tyrosine kinase 2 beta; PDGF Rβ = Platelet-derived growth factor receptor β; PLC-gamma1 = Phospholipase C-gamma 1; Fgr = Gardner-Rasheed feline sarcoma viral; LPS = lipopolysaccharide; ZnO = zinc oxide; FRSM = fermented rapeseed extract; ZnO + LPS = zinc oxide+lipopolysaccharide; FRSM + LPS = fermented rapeseed extract+lipopolysaccharide. (B) Co-culture cells were incubated with FRSM extract (1/50 dilution) and ZnO (50 µM), challenged after 4 h with 1 µg/mL LPS and further incubated for another 24 h. Results are expressed as means ± standard deviation (SD) and standard error (SEM). Values within a row with unlike superscript letters were significantly different (*p* < 0.050). Control = untreated control cells; LPS = cells treated with 1 µg/mL LPS; ZnO = cells treated with ZnO (50 µM); ZnO + LPS = cells treated with ZnO 4 h + LPS for another 24 h; FRSM = cells treated with FRSM (1/50); FRSM + LPS = cells treated with FRSM 4 h + LPS for another 24 h. RSK1/2/ = Ribosomal s6 kinase family; ERK1/2 = Extracellular signal-regulated kinases; Chk-2 = Checkpoint kinase 2; p53 = phospho protein p53; EGF R = epidermal growth factor receptor; p70S6k = Ribosomal protein S6 kinase beta-1; Akt = Protein kinase B; PRAS40 = proline-rich Akt substrate; eNOS = Endothelial NOS (nitric oxide synthase 3); Yes = Associated Transcriptional Regulator; GSK3α/β = Glycogen synthase kinase 3; Lyn = Tyrosine-protein kinase Lyn; LPS = lipopolysaccharide; ZnO = zinc oxide; FRSM = fermented rapeseed extract; ZnO + LPS = zinc oxide+lipopolysaccharide; FRSM + LPS = fermented rapeseed extract + lipopolysaccharide.

## Data Availability

Data are the properties of INCDBNA-IBNA Balotesti and are available upon request to the authors.

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
