# Peer review of "Yeast-Fermented Rapeseed Meal Extract Is Able to Reduce Inflammation and Oxidative Stress Caused by Escherichia coli Lipopolysaccharides and to Replace ZnO in Caco-2/HTX29 Co-Culture Cells"

_ijms, 2022, doi:10.3390/ijms231911640_

Round 1

Reviewer 1 Report

This manuscript presents preliminary in vitro finding regarding the potential of rapeseed meal (an agro-industrial byproduct fermented with yeast S. boulardi) to replace zinc oxide, given the planned banning of the latter in EU by 2023 due to its negative impact on the environment and the development of antibacterial resistance. The lead author and the second co-author have an extensive expertise in this field. This is a very well written review supported by appropriate figures-data, and an adequate and extensive coverage of the related literature. However, the text needs careful editing.

Author Response

We thank very much to reviewer 1 for the valuable review.

As recommended the entire manuscript was read once again and , the text was corrected.

Reviewer 2 Report

Remarks to the Author

The manuscript describes Yeast fermented rapeseed meal extract is able to reduce inflam3 mation and oxidative stress caused by E. coli Lipopolysaccha4 ride and to replace ZnO in Caco-2/HTX29 co-culture cells.  The study is interesting and carried out in detail.

However, manuscript should be carefully revised for typographical and grammatical errors, for example

Line 12“Its use as medicinal will be banned until 2023 in EU, 12 because of environmental negative impact and antibacterial resistance development. This sentence needs to be rewritten

Line 313 alternative compounds to in-feed ZnO with similarly effect.

And many more throughout the manuscript.

Abstract : authors are advised to rewrite it.

Introduction: needs to be rewritten

Scientific name has to be in italics or underlined throughout manuscript.

What is the co-culture medium comprising of?

What cell confluency does 0.1x106 cells/ml indicate? How was it measured?

Can authors discuss how they reduced the possibility of apoptosis by trypsin as authors have not mentioned the protocol for detachment by trypsin?

Overall, I feel the manuscript needs a thorough  typo and grammatical correction before it can be again reviewed for publishing.

Author Response

We thank very much to reviewer 2 for the pertinent observations.

Reviewer #2 Comments:

1)

-comment: Line 12“Its use as medicinal will be banned until 2023 in EU because of environmental negative impact and antibacterial resistance development. This sentence needs to be rewritten

The correction was done in the new version of revised manuscript.

2)

-comment:

Line 313 alternative compounds to in-feed ZnO with similarly effect.

And many more throughout the manuscript.

The correction was done in the new version of revised manuscript.

3)

-comment:

Abstract: authors are advised to rewrite it.

As recommended the abstract was revised and rewritten

4)

-comment:

Introduction: needs to be rewritten

As suggested introduction was revised and rewritten

5)

-comment:

Scientific name has to be in italics or underlined throughout manuscript.

 As recommended the scientific names were written in italics.

6)

-comment:

What is the co-culture medium comprising of?

The co-culture medium was MEM (sigma Aldrich) whose formula contains inorganic salts, amino acids, vitamins and others (glucose, Phenol Red • Na). It was supplemented with 10% foetal bovine serum, 1% (v/v) non-essential amino acids, 2 mM L-glutamine and 1% antibiotic-anti-mycotic solution.

7)

-comment:

What cell confluency does 0.1x106 cells/ml indicate? How was it measured?

Cells were microscopically visualized daily and when the cells covered the entire surface of the well and formed a monolayer, we considered that they had reached the confluence. 

8)

-comment:

Can authors discuss how they reduced the possibility of apoptosis by trypsin as authors have not mentioned the protocol for detachment by trypsin?

Indeed, cells from all the treatments were detached with ethylenediaminetetraacetic acid (EDTA)-trypsin. Cells were first wash in PBS and exposure to trypsin was carefully monitored by reducing the exposure time as much as possible. Trypsin detachment was applied to all treatments including control (untreated cells) and the flow cytometry showed that 88.87% of control cells were alive.

9)

-comment:

Overall, I feel the manuscript needs a thorough typo and grammatical correction before it can be again reviewed for publishing.

The entire manuscript was read again and grammatical corrections were done.

Reviewer 3 Report

Ref: Ijms-1907972

Journal: International Journal of Molecular Sciences

Dear authors;

The manuscript entitled “Yeast fermented rapeseed meal extract is able to reduce inflammation and oxidative stress caused by Escherichia coli Lipopolysaccharide and to replace ZnO in Caco-2/HTX29 co-culture cells” has been reviewed. In the current manuscript, you tested in vitro the effect of a fermented extract of rapeseed meal as a substitute for ZnO. The subject is very interesting, you have followed the right scientific way in each experiment and the manuscript is well organised. However, the manuscript needs to be improved in terms of methodology and results. I recommend revisions before the publication of the paper in International Journal of Molecular Sciences. Consequently, I would like to raise some comments, and recommendations about the manuscript:

1. In the title, the full name of Escherichia coli should be indicated.

2. Abstract and introduction generally are good with only a few grammatical errors.

3. In the results section, subsection “ROS production”, figure x doesn't exist in the manuscript, I think it's figure 2.

4. In the results section, subsection “Protein oxidation”, figure xx doesn't exist in the manuscript, I think it's figure 3.

5. For the numerical results given in the manuscript text, I suggest that the authors also add the standard deviation.

6. For figure 4 and 5, the results of statistical analyses should be added.

7. For table 1, the results of statistical analyses and standard deviation should be added.

8. For table 2, 3 and 4 standard deviations should be added.

9. The discussion section generally is good. But All species names should be in italics. Please use italics for all species names, especially those in the discussion section.

10. Materials and Methods section, in “Fermented rapeseed meal extract and ZnO solutions” subsection, indicates the water type that has been homogenised with the rapeseed meal.

11. In most sub-sections the references used are missing. Please, add the reference used.

12. Please, add a conclusion section, in which you can highlight the importance of the results obtained

13. The references need to be improved by using recent references.

Best regards

Author Response

List of itemized changes made in response to points raised by the reviewer 2

We thank very much to the reviewer for the pertinent observations.

Reviewer #2 Comments:

Dear authors;

The manuscript entitled “Yeast fermented rapeseed meal extract is able to reduce inflammation and oxidative stress caused by Escherichia coli Lipopolysaccharide and to replace ZnO in Caco-2/HTX29 co-culture cells” has been reviewed. In the current manuscript, you tested in vitro the effect of a fermented extract of rapeseed meal as a substitute for ZnO. The subject is very interesting, you have followed the right scientific way in each experiment and the manuscript is well organised. However, the manuscript needs to be improved in terms of methodology and results. I recommend revisions before the publication of the paper in International Journal of Molecular Sciences. Consequently, I would like to raise some comments, and recommendations about the manuscript:

1)

-comment:

 In the title, the full name of Escherichia coli should be indicated.

The correction was done in the new version of revised manuscript.

2)

-comment:

Abstract and introduction generally are good with only a few grammatical errors.

As recommended the abstract and introduction were carefully read again and the grammatical errors were corrected.

3)

-comment:

In the results section, subsection “ROS production”, figure x doesn't exist in the manuscript, I think it's figure 2.

Deeply sorry for this error. The correction was done in the new version of the manuscript.

4)

-comment:

In the results section, subsection “Protein oxidation”, figure xx doesn't exist in the manuscript, I think it's figure 3.

Deeply sorry for this error. The correction was done in the new version of the manuscript.

5)

-comment:

 For the numerical results given in the manuscript text, I suggest that the authors also add the standard deviation.

As suggested standard deviation was added for all numerical values in the tables.

6)

 -comment:

For figure 4 and 5, the results of statistical analyses should be added.

Deeply sorry for this error. Figure 4 and 5 were revised and statistical significance was added.

7) For table 1, the results of statistical analyses and standard deviation should be added.

As suggested standard deviation was added for all results in the table 1.

8) For table 2, 3 and 4 standard deviations should be added.

As suggested standard deviation was added for all results in the table 2, 3 and 4.

9) The discussion section generally is good. But All species names should be in italics. Please use italics for all species names, especially those in the discussion section.

The correction was done. All species names are in italics in the new version of the manuscript.

10) Materials and Methods section, in “Fermented rapeseed meal extract and ZnO solutions” subsection, indicates the water type that has been homogenised with the rapeseed meal.

As recommended the type of water used for homogenisation was added in material and methods.

11) In most sub-sections the references used are missing. Please, add the reference used.

As recommended the missing references was added.

12) Please, add a conclusion section, in which you can highlight the importance of the results obtained

A conclusion was added in the new version of the manuscript.

13) The references need to be improved by using recent references.

New references were added in the new version of the new manuscript.

Round 2

Reviewer 2 Report

Manuscript still needs little more editing for language and prepositions etc

Reviewer 3 Report

Manuscript Number: ijms-1907972R1

Journal: International Journal of Molecular Sciences

Dear editor;

The revised version of the study titled "Yeast fermented rapeseed meal extract is able to reduce inflammation and oxidative stress caused by Escherichia coli Lipopolysaccharide and to replace ZnO in Caco-2/HTX29 co-culture cells" has been reviewed. In my opinion, the authors have satisfactorily addressed all my previous issues. The paper can be accepted and published in its current form.

With best regards